# Modular Universal Reparameterization:
# Deep Multi-task Learning Across Diverse Domains

**Elliot Meyerson**
Cognizant
elliot.meyerson@cognizant.com

**Risto Miikkulainen**[1,2]
Cognizant[1]
The University of Texas at Austin[2]
risto@cs.utexas.edu

## Abstract

As deep learning applications continue to become more diverse, an interesting question arises: Can general problem solving arise from jointly learning several such diverse tasks? To approach this question, deep multi-task learning is extended in this paper to the setting where there is no obvious overlap between task architectures. The idea is that any set of (architecture,task) pairs can be decomposed into a set of potentially related subproblems, whose sharing is optimized by an efficient stochastic algorithm. The approach is first validated in a classic synthetic multi-task learning benchmark, and then applied to sharing across disparate architectures for vision, NLP, and genomics tasks. It discovers regularities across these domains, encodes them into sharable modules, and combines these modules systematically to improve performance in the individual tasks. The results confirm that sharing learned functionality across diverse domains and architectures is indeed beneficial, thus establishing a key ingredient for general problem solving in the future.

## 1 Introduction

Deep learning methods and applications continue to become more diverse. They now solve problems that deal with fundamentally different kinds of data, including those of human behavior, such as vision, language, and speech, as well as those of natural phenomena, such as biological, geological, and astronomical processes.

Across these domains, deep learning architectures are painstakingly customized to different problems. However, despite this extreme customization, a crucial amount of functionality is shared across solutions. For one, architectures are all made of the same ingredients: some creative composition and concatenation of high-dimensional linear maps and elementwise nonlinearities. They also share a common set of training techniques, including popular initialization schemes and gradient-based optimization methods. The fact that the same small toolset is successfully applied to all these problems implies that the problems have a lot in common. Sharing these tools across problems exploits some of these commonalities, i.e., by setting a strong prior on the kinds of methods that will work. Such sharing is *methodological*, with humans determining what is shared.

This observation begs the question: Are there commonalities across these domains that methodological sharing cannot capture? Note that this question is different from that addressed by previous work in deep multi-task learning (DMTL), where the idea is to share knowledge across tasks in the same domain or modality, such as within vision [5, 30, 33, 39, 57, 61] or language [9, 13, 16, 31, 34]. In contrast, this question is fundamental to general problem solving: Can it be beneficial to share *learned* functionality across a diverse set of tasks, such as a 2D convolutional vision network, an LSTM model for natural language, and a 1D convolutional model for genomics? Specifically, this paper considers the following problem: Given an arbitrary set of (architecture,task) pairs, can learned functionality be shared across architectures to improve performance in each task?

Drawing on existing approaches to DMTL, a first approach to this problem is developed, showing that such effective sharing is indeed possible. The approach is based on decomposing the general multi-task learning problem into several fine-grained and equally-sized subproblems, or *pseudo-tasks*. Training a set of (architecture,task) pairs then corresponds to solving a set of related pseudo-tasks, whose relationships can be exploited by shared functional modules. To make this framework practical, an efficient search algorithm is introduced for optimizing the mapping between pseudo-tasks and the modules that solve them, while simultaneously training the modules themselves. The approach, modular universal reparameterization (MUiR), is validated in a synthetic MTL benchmark problem, and then applied to large-scale sharing between the disparate modalities of vision, NLP, and genomics. It leads to improved performance on each task, and highly-structured architecture-dependent sharing dynamics, in which the modules that are shared more demonstrate increased properties of generality. These results show that MUiR makes it possible to share knowledge across diverse domains, thus establishing a key ingredient for building general problem solving systems in the future.

## 2   Problem Statement and Related Work

This paper is concerned with the following question: *Given an arbitrary set of (architecture,task) pairs, can learned functionality be shared across architectures to improve performance in each task?* Any method that answers this question must satisfy two requirements: (1) It must support any given set of architectures, and (2) it must *align* parameters across the given architectures.

Parameters in two architectures are *aligned* if they have some learnable tensor in common. An alignment across architectures implies how tasks are related, and how much they are related. The goal of DMTL is to improve performance across tasks through joint training of aligned architectures, exploiting inter-task regularities. In recent years, DMTL has been applied within areas such as vision [5, 30, 33, 39, 57, 61], natural language [9, 13, 16, 31, 34], speech [19, 46, 55], and reinforcement learning [11, 20, 51]. The rest of this section reviews existing DMTL methods, showing that none of these methods satisfy both conditions (1) and (2).

The classical approach to DMTL considers a joint model across tasks in which some aligned layers are shared completely across tasks, and the remaining layers remain task-specific [7]. In practice, the most common approach is to share all layers except for the final classification layers [11, 13, 18, 19, 20, 31, 42, 55, 61]. A more flexible approach is to not share parameters exactly across shared layers, but to factorize layer parameters into shared and task-specific factors [3, 23, 28, 32, 44, 56, 57]. Such approaches work for any set of architectures that have a known set of aligned layers. However, these methods only apply when such alignment is known *a priori*. That is, they do not meet condition (2).

One approach to overcome the alignment problem is to design an entirely new architecture that integrates information from different tasks and is maximally shared across tasks [5, 16, 22]. Such an approach can even be used to share knowledge across disparate modalities [22]. However, by disregarding task-specific architectures, this approach does not meet condition (1). Related approaches attempts to learn how to assemble a set of shared modules in different ways to solve different tasks, whether by gradient descent [37], reinforcement learning [45], or evolutionary architecture search [30]. These methods also construct new architectures, so they do not meet condition (1); however, they have shown that including a small number of location-specific parameters is crucial to sharing functionality across diverse locations.

Drawing on the methods above, this paper introduces a first approach that meets both conditions. First, a simple decomposition is introduced that applies to any set of architectures and supports automatic alignment. This decomposition is extended to include a small number of location-specific parameters, which are integrated in a manner mirroring factorization approaches. Then, an efficient alignment method is developed that draws on automatic assembly methods. These methods combine to make it possible to share effectively across diverse architectures and modalities.

## 3   Modular Universal Reparameterization

This section presents a framework for decomposing sets of (architecture,task) pairs into equally-sized subproblems (i.e., pseudo-tasks), sharing functionality across aligned subproblems via a simple factorization, and optimizing this alignment with an efficient stochastic algorithm.

## 3.1 Decomposition into linear pseudo-tasks

Consider a set of $T$ tasks $\{\{\boldsymbol{x}_{ti}, \boldsymbol{y}_{ti}\}_{i=1}^{N_t}\}_{t=1}^{T}$, with corresponding model architectures $\{\mathcal{M}_t\}_{t=1}^{T}$, each parameterized by a set of trainable tensors $\theta_{\mathcal{M}_t}$. In MTL, these sets have non-trivial pairwise intersections, and are trained in a joint model to find optimal parameters $\theta_{\mathcal{M}_t}^{\star}$ for each task:

$$\bigcup_{t=1}^{T} \theta_{\mathcal{M}_t}^{\star} = \underset{\bigcup_{t=1}^{T} \theta_{\mathcal{M}_t}}{\operatorname{argmin}} \frac{1}{T} \sum_{t=1}^{T} \frac{1}{N_t} \sum_{i=1}^{N_t} \mathcal{L}_t(\boldsymbol{y}_{ti}, \hat{\boldsymbol{y}}_{ti}), \tag{1}$$

where $\hat{\boldsymbol{y}}_{ti} = \mathcal{M}_t(\boldsymbol{x}_{ti}; \theta_{\mathcal{M}_t})$ is a prediction and $\mathcal{L}_t$ is a sample-wise loss function for the $t$th task. Given fixed task architectures, the key question in designing an MTL model is how the $\theta_{\mathcal{M}_t}$ should be aligned. The following decomposition provides a generic way to frame this question.

Suppose each tensor in each $\theta_{\mathcal{M}_t}$ can be decomposed into equally-sized parameter blocks $\boldsymbol{B}_\ell$ of size $m \times n$, and there are $L$ such blocks total across all $\theta_{\mathcal{M}_t}$. Then, the parameterization for the entire joint model can be rewritten as:

$$\bigcup_{t=1}^{T} \theta_{\mathcal{M}_t} = (\boldsymbol{B}_1, \dots, \boldsymbol{B}_L). \tag{2}$$

That is, the entire joint parameter set can be regarded as a single tensor $\boldsymbol{B} \in \mathbb{R}^{L \times m \times n}$. The vast majority of parameter tensors in practice can be decomposed in this way such that each $\boldsymbol{B}_\ell$ defines a linear map. For one, the $pm \times qn$ weight matrix of a dense layer with $pm$ inputs and $qn$ outputs can be broken into $pq$ blocks of size $m \times n$, where the $(i, j)$th block defines a map between units $im$ to $(i+1)m - 1$ of the input space and units $jn$ to $(j+1)n - 1$ of the output space. This approach can be extended to convolutional layers by separately decomposing each matrix corresponding to a single location in the receptive field. Similarly, the parameters of an LSTM layer are contained in four matrices, each of which can be separately decomposed. When $m$ and $n$ are relatively small, the requirement that $m$ and $n$ divide their respective dimensions is a minor constraint; layer sizes can be adjusted without noticeable effect, or overflowing parameters from edge blocks can be discarded.

Now, if each $\boldsymbol{B}_\ell$ defines a linear map, then training $\boldsymbol{B}$ corresponds to solving $L$ linear *pseudo-tasks* [38] that define subproblems within the joint model. Suppose $\boldsymbol{B}_\ell$ defines a linear map in $\mathcal{M}_t$. Then, the $\ell$th pseudo-task is solved by completing the computational graph of $\mathcal{M}_t$ with the subgraph corresponding to $\boldsymbol{B}_\ell$ removed. The $\ell$th pseudo-task is denoted by a five-tuple

$$(\mathcal{E}_\ell, \theta_{\mathcal{E}_\ell}, \mathcal{D}_\ell, \theta_{\mathcal{D}_\ell}, \{\boldsymbol{x}_{ti}, \boldsymbol{y}_{ti}\}_{i=1}^{N_t}), \tag{3}$$

where $\mathcal{E}_\ell$ is the encoder that maps each $\boldsymbol{x}_{ti}$ to the input of a function solving the pseudo-task, and $\mathcal{D}_\ell$ takes the output of that function (and possibly $\boldsymbol{x}_{ti}$) to the prediction $\hat{\boldsymbol{y}}_{ti}$. The parameters $\theta_{\mathcal{E}_\ell}$ and $\theta_{\mathcal{D}_\ell}$ characterize $\mathcal{E}_\ell$ and $\mathcal{D}_\ell$, respectively.

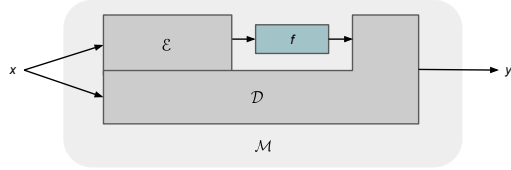

Figure 1: *Pseudo-task decomposition*. Architecture $\mathcal{M}$, for task $\{\boldsymbol{x}_i, \boldsymbol{y}_i\}_{i=1}^{N}$, induces a pseudo-task solved by a function $f$. $\mathcal{E}$ is an encoder that provides input to $f$, and $\mathcal{D}$ is a decoder that uses the output of $f$ to produce the final prediction. If $f$ is effective for many [task, encoder, decoder] combinations, then it shows generic functionality.

In general, given a pseudo-task, the model for the $t$th task is completed by a differentiable function $f$ that connects the pseudo-task's inputs to its outputs. The goal for solving this pseudo-task is to find a function that minimizes the loss of the underlying task. The completed model is given by

$$\hat{\boldsymbol{y}}_t = \mathcal{D}_\ell(f(\mathcal{E}_\ell(\boldsymbol{x}_t; \theta_{\mathcal{E}_\ell}); \theta_f), \boldsymbol{x}_t; \theta_{\mathcal{D}_\ell}). \tag{4}$$

This formulation is depicted in Figure 1. Since all $L$ pseudo-tasks induced by Eq. 2 have the same input-output specification, if $f$ solves one of them, it can be applied to any of them in a modular way.

Since all pseudo-tasks are derived from the same universe of tasks and architectures, sharing modules across them can be valuable. Indeed, sharing across related parameter blocks is a common tool to improve generalization in deep learning. For example, a convolutional layer can be viewed as a dense layer with parameter blocks shared across space, and a recurrent layer as a sequential network of dense layers with parameter blocks shared across depths, i.e., time. Similarly, the standard DMTL approach is to design a joint architecture with some parameter blocks shared across related tasks. This paper extends DMTL to sharing factors across related pseudo-tasks.

## 3.2 Reparameterization by hypermodules

Assuming an effective alignment of related pseudo-tasks exists, how should parameters be shared across them? Reusing modules at qualitatively different locations in a network has been successful when a small number of location-specific parameters are included to increase flexibility [30, 37], and has been detrimental when such parameters are not included [45]. To include such parameters in a simple and flexible way, and avoid additional assumptions about the kind of sharing that can occur, each $B_\ell$ can be generated by a *hypermodule*, the module-specific analog of a hypernetwork [15, 48].

Associate with the $\ell$th pseudo-task a context vector $z_\ell \in \mathbb{R}^c$. Suppose there is also a collection of $K$ hypermodules $\{H_k\}_{k=1}^K$, with $H_k \in \mathbb{R}^{c \times m \times n}$, and let $\psi : \{1, \dots, L\} \to \{H_k\}_{k=1}^K$ be an alignment function that indicates which hypermodule solves the $\ell$th pseudo-task. Then, the parameters of the underlying architectures are generated by

$$B_\ell = \psi(\ell) \; \bar{\times}_1 \; z_\ell, \tag{5}$$

where $\bar{\times}_1$ denotes the *1-mode (vector) product* of a tensor and a vector [25]. In other words, the value at $B_{\ell ij}$ is the dot product between $z_\ell$ and the fiber in $\psi(\ell)$ associated with the $(i, j)$th element of $B_\ell$. With the additional goal of optimizing $\psi$, the block decomposition (Eq. 2) can now be written as

$$\bigcup_{t=1}^T \theta_{\mathcal{M}_t} = [(H_1, \dots, H_K), (z_1, \dots, z_L)]. \tag{6}$$

To accurately apply Eq. 6 to a set of architectures, the parameter initialization scheme must be preserved. Say the parameters of a layer are initialized i.i.d. with variance $\sigma^2$ and mean 0, and each $B_\ell$ is initialized with a distinct hypermodule $\psi(\ell) = H_\ell$. When $c > 1$, $B_{\ell ij} = \langle H_{\ell:ij}, z_\ell \rangle$ is a sum of random variables, so it is impossible to initialize $H_\ell$ and $z_\ell$ i.i.d. such that $B_{\ell ij}$ is initialized from a uniform distribution. However, it is possible to initialize $B_{\ell ij}$ from a normal distribution, by initializing $H_\ell$ from a normal distribution $\mathcal{N}(0, \sigma_H^2)$ and initializing $z_\ell$ with constant magnitude $|z|$:

$$B_{\ell ij} = \langle H_{\ell:ij}, z_\ell \rangle \sim c|z|\mathcal{N}(0, \sigma_H^2) = \mathcal{N}(0, z^2 c^2 \sigma_H^2) = \mathcal{N}(0, \sigma^2) \implies |z| = \frac{\sigma}{c\sigma_H}. \tag{7}$$

In this paper, $\sigma^2$ and $\sigma_H^2$ are determined by He normal initialization [17], which implies a unique $|z|$. Although $z_\ell$ could be initialized uniformly from $\{-z, z\}^c$, it is instead initialized to the constant $z$, to encourage compatibility of hypermodules across contexts. Similarly, the fact that all $H_k$ have the same $\sigma_H^2$ makes it easier for them to capture functionality that applies across pseudo-tasks.

Although it is pessimistic to initialize each pseudo-task with its own hypermodule, parsimonious models can be achieved through optimization of $\psi$. Using the same hypermodule for many pseudo-tasks has the side-benefit of reducing the size of the joint model. The original model in Eq. 2 has $Lmn$ trainable parameters, while Eq. 6 has $Lc + Kcmn$, which is more parsimonious only when $K < \frac{L(mn-c)}{cmn} < \frac{L}{c}$, i.e., when each hypermodule is used for more than $c$ pseudo-tasks on average. However, after training, any hypermodule used fewer than $c$ times can be replaced with the parameters it generates, so the model complexity at inference is never greater than that of the original model: $(L - L_o)c + Kcmn + L_omn \leq Lmn$, where $L_o$ is the number of pseudo-tasks parameterized by hypermodules used fewer than $c$ times. An algorithm that improves parsimony in this way while exploiting related pseudo-tasks is introduced next.

## 3.3 Interleaved optimization of pseudo-task alignment

Given the above decomposition and reparameterization, the goal is to find an optimal alignment $\psi$, given by a fixed-length mapping $(\psi(1), \dots, \psi(L))$, with $K$ possible choices for each element. Let $h$ be a scoring function that returns the performance of a mapping via training and evaluation of the joint model. In order to avoid training the model from scratch each iteration, existing DMTL approaches that include nondifferentiable optimization interleave this optimization with gradient-based updates [8, 30, 33, 38, 45]. These methods take advantage of the fact that at every iteration there are $T$ scores, one for each task. These scores can be optimized in parallel, and faster convergence is achieved, by effectively decomposing the problem into $T$ subproblems. This section illustrates that such problem decomposition can be greatly expanded, leading to practical optimization of $\psi$.

| Decomposition Level | None (Multi-task) | Per-task (Single-task) | Per-block (Pseudo-task) |
|---|---|---|---|
| Expected Convergence Time | $O(KL \log L)$ | $O\left(\frac{KL(\log L - \log T) \log T}{T}\right)$ | $O(K \log L)$ |

Table 1: *Complexity of pseudo-task alignment.* This table gives the expected times of Algorithm 1 for finding the optimal mapping of $L$ pseudo-tasks to $K$ hypermodules, in a model with $T$ tasks. The runtime of pseudo-task-level optimization scales logarithmically with the size of the model.

In general, $\psi$ may be decomposed into $D$ submappings $\{\psi_d\}_{d=1}^{D}$, each with a distinct evaluation function $h_d$. For simplicity, let each submapping be optimized with an instance of the $(1+\lambda)$-EA, a Markovian algorithm that is robust to noise, dynamic environments, and local optima [12, 40, 49], and is a component of existing DMTL methods [30, 38]. The algorithm generates new solutions by resampling elements of the best solution with an optimal fixed probability. Algorithm 1 extends the $(1+\lambda)$-

---

**Algorithm 1** Decomposed $K$-valued $(1 + \lambda)$-EA

1: Create initial solutions $\psi_1^0, \ldots, \psi_D^0$ each of length $\frac{L}{D}$
2: **while** any $\psi_d^0$ is suboptimal **do**
3:     **for** $d = 1$ **to** $D$ **do**
4:        **for** $i = 1$ **to** $\lambda$ **do**
5:           $\psi_d^i \leftarrow \psi_d^0$
6:           **for** $\ell = 1$ **to** $\frac{L}{D}$ **do**
7:              With probability $\frac{D}{L}$, $\psi_d^i(\ell) \sim \mathcal{U}(\{\boldsymbol{H}_k\}_{k=1}^{K})$
8:        **for** $t = 1$ **to** $d$ **do**
9:           $\psi_d^0 = \text{argmax}_{\psi_d^i} \, h(\psi_d^i)$

---

EA to optimizing submappings in parallel. Assume each $\psi_d$ has length $L/D$, $\lambda = 1$, all $h_d$ are linear, i.e., $h_d(\psi_d) = \sum_{\ell=1}^{L} w_{d\ell} \cdot I(\psi_d(\ell) = \psi_d^\star(\ell))$, where $w_{d\ell}$ are positive scalars, $I$ is the indicator function, and $\psi^\star$ is a unique optimal mapping, with $\psi^\star(\ell) = \boldsymbol{H}_1 \, \forall \ell$. The runtime of this algorithm (number of iterations through the while loop) is summarized by the following result (proof in S.1):

**Theorem 3.1.** *The expected time of the decomposed $K$-valued (1+1)-EA is $O\left(\frac{KL(\log L - \log D) \log D}{D}\right)$, when all $h_d$ are linear.*

Resulting runtimes for key values of $D$ are given in Table 1. As expected, setting $D = T$ gives a substantial speed-up over $D = 1$. However, when $T$ is small relative to $L$, e.g., when sharing across a small number of complex models, the factor of $L$ in the numerator is a bottleneck. Setting $D = L$ overcomes this issue, and corresponds to having a distinct evaluation function for each pseudo-task.

The pessimistic initialization suggested in Section 3.2 avoids initial detrimental sharing, but introduces another bottleneck: large $K$. This bottleneck can be overcome by sampling hypermodules in Line 7 proportional to their usage in $\psi^0$. Such proportional sampling encodes a prior which biases search towards modules that already show generality, and yields the following result (proof in S.2):

**Theorem 3.2.** *The expected time of the decomposed $K$-valued (1+1)-EA with pessimistic initialization and proportional sampling is $O(\log L)$, when $D = L$, and all $h_d$ are linear.*

Again, this fast convergence requires a pseudo-task-level evaluation function $h$. The solution adopted in this paper is to have the model indicate its hypermodule preference directly through backpropagation, by learning a softmax distribution over modules at each location. Similar distributions over modules have been learned in previous work [30, 37, 47]. In Algorithm 1, at a given time there are $1 + \lambda$ active mapping functions $\{\psi^i\}_{i=0}^{\lambda}$. Through backpropagation, the modules $\{\psi^i(\ell)\}_{i=0}^{\lambda}$ for each location $\ell$ can compete by generalizing Eq. 5 to include a soft-merge operation:

$$\boldsymbol{B}_\ell = \sum_{i=0}^{\lambda} \psi^i(\ell) \,\bar{\times}_1\, \boldsymbol{z}_\ell \cdot \text{softmax}(\boldsymbol{s}_\ell)_i, \tag{8}$$

where $\boldsymbol{s}_\ell \in \mathbb{R}^{\lambda+1}$ is a vector of weights that induces a probability distribution over hypermodules. Through training, the learned probability of $\text{softmax}(\boldsymbol{s}_\ell)_i$ is the model's belief that $\psi^i(\ell)$ is the best option for location $\ell$ out of $\{\psi^i(\ell)\}_{i=0}^{\lambda}$. Using this belief function, Algorithm 1 can optimize $\psi$ while simultaneously learning the model parameters. Each iteration, the algorithm trains the model via Eq. 8 with backpropagation for $n_{\text{iter}}$ steps, and $h(\psi_\ell^i)$ returns $\sum_{j=0}^{\lambda} \text{softmax}(\boldsymbol{s}_\ell)_j \cdot I(\psi_\ell^j = \psi_\ell^i)$, accounting for duplicates. In contrast to existing model-design methods, task performance does not guide search; this avoids overfitting to the validation set over many generations. Validation performance is only used for early stopping. Pseudocode for the end-to-end algorithm, along with additional training considerations, are given in S.3. The algorithm is evaluated experimentally in the next section.

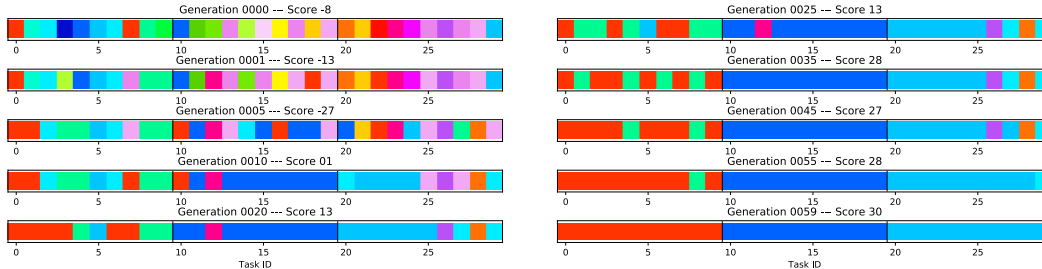

Figure 2: *Visualizing convergence.* These images show the convergence of $\psi$ on the synthetic dataset. Each color corresponds to a distinct hypermodule. The color shown at each location is the hypermodule currently in use for that task. After generation 59 the model remains at the optimal solution indefinitely, demonstrating the efficient convergence of MUiR.

The theoretical scalability of the algorithm means it can be applied in settings where existing DMTL module assembly methods are infeasible. For instance, when learning the alignment with soft ordering [37] the module operations increase quadratically; sampling from the softmax instead [47] would require thousands of additional parameters per module location; learning the alignment with CTR [30] is infeasibly complex via Theorem 3.1. These limitations are highlighted in the fact that experiments with existing approaches use at most 4 [37], 4 [30], and 10 [45] modules, i.e., orders of magnitude fewer than what is considered in this paper (e.g., more than 10K modules in Section 4.2).

# 4 Experiments

This section evaluates the approach developed in Section 3. First, the dynamics of the approach are validated a synthetic MTL benchmark. Second, the approach is applied to a scale-up problem of sharing across diverse architectures and modalities. See S.4 for additional experimental details.

## 4.1 Validating framework dynamics on a synthetic dataset

This section considers an MTL problem where the ground truth alignment is known. The dataset contains three groups of ten linear regression tasks with input dimension 20, but only 15 training samples per task [23]. The ground truth parameter vector for tasks within a group differ only by a scalar. Tasks cannot be solved without exploiting this regularity. Two versions of the problem were considered, one with Gaussian noise added to sample outputs, and one with no noise. As in previous work, each task model is linear, consisting of a single weight vector $\in \mathbb{R}^{20}$. In the single-task (STL) case, these vectors are trained independently. In the MTL case (MUiR), $c = 1$, and each task is reparameterized with a single hypermodule $\in \mathbb{R}^{1 \times 20 \times 1}$. So, Algorithm 1 is initialized with 30 hypermodules, and should converge to using only three, i.e., one for each group. For comparison, a Random search setup is included (i.e., replacing argmax in Algorithm 1 with a random choice), as well as an Oracle setup, in which $\psi$ is fixed to the true group alignment. Unlike in previous work, five training samples for each task were withheld as validation data, making the setup more difficult.

| Method | Clean | Noisy |
|---|---|---|
| STL [23] | - | 0.97 |
| MTL-FEAT [3] | - | 0.48 |
| DG-MTL [23] | - | 0.42 |
| GO-MTL [28] | - | **0.35** |
| STL (ours) | $1.35 \pm 0.01$ | $1.49 \pm 0.01$ |
| MUiR + Random | $1.26 \pm 0.04$ | $4.67 \pm 1.48$ |
| MUiR + Oracle | $0.77 \pm 0.77$ | $\mathbf{0.37} \pm 0.00$ |
| MUiR + Optimization | $\mathbf{0.00} \pm 0.00$ | $\mathbf{0.38} \pm 0.00$ |

Table 2: *Synthetic results.* MUiR achieves perfect test RMSE in the clean case, even outperforming the Oracle, which can sometimes overfit. MUiR similarly outperforms baselines in the noisy case. Since a linear model is optimal for this dataset, MUiR cannot improve over the best linear method, but it achieves comparable results despite differences in the setup that make it more difficult: withholding data for validation and absence of additional regularization. Also, in contrast to the other methods, MUiR learns the number of groups automatically.

MUiR quickly converges to the true underlying grouping in the noiseless case (Figure 2), and yields optimal test loss (Table 2). In the noisy case, MUiR results in a similar improvement over the baselines. Since a linear model is optimal for this dataset, MUiR cannot improve over the best linear method, but it achieves comparable results, despite differences in the setup that make generalization more difficult: withholding data for validation and absence of additional regularization. These results

| Modality | Architecture | Baseline | Intratask | W+S | W+D | S+D | W+S+D | L+S | L+D | L+S+D |
|---|---|---|---|---|---|---|---|---|---|---|
| Vision | WRN-40-1 (W) | **8.48** | **8.50** | 8.69 | 9.20 | - | 9.02 | - | - | - |
| Text | Stacked LSTM (S) | 134.41 | 132.06 | 130.63 | - | 132.62 | **128.10** | **129.73** | - | 130.77 |
| DNA | DeepBind-256 (D) | 0.1540 | 0.1466 | - | **0.1461** | 0.1469 | **0.1464** | - | 0.1469 | **0.1464** |
| Vision | LeNet (L) | 21.08 | 20.67 | - | - | - | - | 21.02 | **19.59** | **20.23** |

Table 3: *Cross-modal results.* This table shows the performance of each architecture across a chain of comparisons. *Baseline* trains the underlying model; *Intratask* uses MUiR with a single task architecture; the remaining setups indicate multiple architectures trained jointly with MUiR. Lower scores are better: classification error for vision, perplexity for text and MSE for DNA. For each architecture, the top two setups are in bold. The LSTM, DeepBind, and LeNet models all benefit from cross-modal sharing; and in all 16 cases, MUiR improves their performance over Baseline. Although the text and DNA models both benefit from sharing with WRN, the effect is not reciprocated. The fact that LeNet improves suggests that it is not a problem in transferring across modalities, but that WRN has an architecture that is easier to share *from* than *to*. Overall, the ability of MUiR to improve performance, even in the intratask case, indicates that it can exploit pseudo-task regularities.

show that the softmax evaluation function effectively determines the value of hypermodules at each location. The next section shows that the algorithm scales to more complex problems.

## 4.2 Sharing across diverse architectures and modalities

This experiment applies MUiR in its intended setting: sharing across diverse architectures and modalities. The hypermodules generate $16 \times 16$ linear maps, and have context size $c = 4$, as in previous work on hypernetworks [15]. The joint model shares across a vision problem, an NLP problem, and a genomics problem (see S.5 for additional dataset and architecture details).

The first task is *CIFAR-10*, the classic image classification benchmark of 60K images [26]. As in previous work on hypernetworks, WideResNet-40-1 (WRN) is the underlying model [15, 58], yielding 2268 blocks to parameterize with hypermodules. The second task is *WikiText-2* language modeling benchmark with over 2M tokens [36]. The underlying model is the standard stacked LSTM model with two LSTM layers each with 256 units [59], yielding 4096 blocks. The third task is *CRISPR binding prediction*, where the goal is to predict the propensity of a CRISPR protein complex to bind to (and cut) unintended locations in the genome [21]. The dataset contains binding affinities for over 30M base pairs. The underlying model, DeepBind-256, is from the DeepBind family of 1D-convolutional models designed for protein binding problems [2, 60], yielding 6400 blocks.

### 4.2.1 Performance comparison across (architecture,task) subsets

For each of these three task-architecture pairs, a chain of comparisons were run, with increasing generality: a Baseline that trained the original architecture; an Intratask setup that applied MUiR optimization within a single task model; cross-modal optimization for each pair of tasks; and a cross-modal run across all three tasks. The main result is that the text and genomics models always improve when they are trained with MUiR, and improve the most when they are trained jointly with the WRN model (Table 3). This result raises a key question: Does the (WRN,vision) pair behave differently because of WRN or because of vision? To answer this question, an additional set of experiments were run using LeNet [29] as the vision model. This model does indeed always improve with MUiR, and improves the most with cross-modal sharing (Table 3), while similarly improving the text and genomics models. The improvements for all three tasks are significant (S.4). Overall, the results confirm that MUiR can improve performance by sharing across diverse modalities. A likely reason that the benefit of WRN is one-directional is that the modules in WRN are highly specialized to work together as a deep stack. They provide useful diversity in the search for general modules, but they are hard to improve using such modules. This result is important because it both illustrates where the power of MUiR is coming from (diversity) and identifies a key challenge for future methods.

### 4.2.2 Analysis of module sharing dynamics

To understand the discovery process of MUiR, Figure 3a shows the number of modules used exclusively by each subset of tasks over time in a W+D+S run. The relative size of each subset stabilizes as

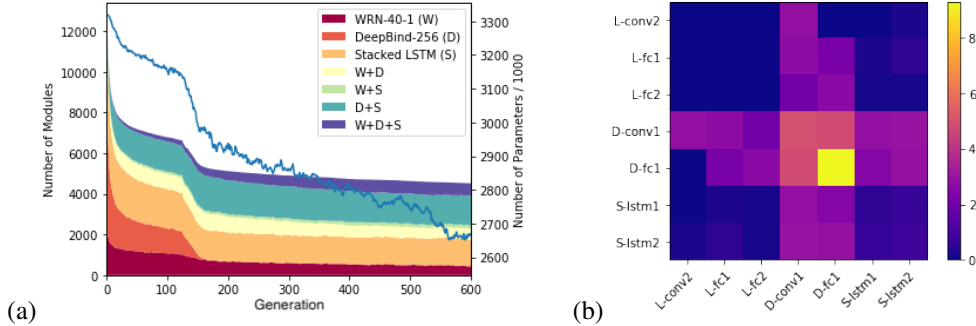

(a)                                                                (b)

Figure 3: (a) *Module sharing over time.* The number of modules shared exclusively by each subset of tasks is shown for a MUiR run. The differences across subsets show that MUiR optimizes alignment in an architecture-dependent way. For example, the number of modules used only by the WRN and LSTM models always stays small, and the number used only by the DeepBind model eventually shrinks to almost zero, suggesting that the genomics model plays a central role in sharing. As a side-benefit of this optimization, the number of parameters in the model decreases (blue line). (b) *Layer-level sharing.* To measure sharing across pairs of layers, for each pair in an L+S+D run, this heatmap shows how many times more likely pairs of pseudo-tasks from those layers are to use the same module than they would by chance. Sharing is highly architecture-dependent, with the 1D-convolutional model playing a central role between the 2D-convolutional and 1D-LSTM models.

$\psi$ is optimized, and is consistent over independent runs, showing that MUiR shares in an architecture-dependent way. In particular, the number of modules used only by W and S models remains small, and the number used only by D shrinks to near zero, suggesting that the genomics model plays a central role in sharing. Analyzed at the layer level in the L+S+D setup, the bulk of sharing does indeed involve D (Figure 3b). D and L are both convolutional, while D and S process 1-dimensional input, which may make it easier for L and S to share with D than directly with each other.

A side-benefit of MUiR is that the number of model parameters decreases over time (up to 20% in Figure 3a), which is helpful when models need to be small, e.g., on mobile devices. Such shrinkage is achieved when the optimized model has many modules that are used for many pseudo-tasks. Hypermodules are considered *generic* if they are used more than $c$ times in the joint model, and *specific* otherwise. Similarly, pseudo-tasks are considered generic if they use generic modules and specific otherwise, along with their contexts and generated linear maps. Sets of generic and specific tensors were compared based on statistical properties of their learned parameters. The generic tensors had significantly smaller average standard deviation, L2-norm, and max value (Table 4). Such a tighter distribution of parameters indicates greater generality [4, 27].

### 4.2.3 Ablations and DMTL comparisons

Even though their application seems unnatural for the cross-domain problem, experiments were performed using existing DMTL methods: *classical* DMTL (e.g., [13, 19, 61]), i.e., where aligned parameters are shared exactly across tasks; and *parallel adapters* [44], which is state-of-the-art for vision MTL. Both of these methods require a hierarchical alignment of parameters across architectures. Here, the most natural hierarchical alignment is used, based on a topological sort of the block locations within each architecture: the $i$th location uses the $i$th parameter block. MUiR outperforms the existing methods (Table 6). Interestingly, the existing methods each outperforms

| Parameter Group | Stdev | Mean | Norm | Max |
|---|---|---|---|---|
| Hypermodules | 7e-4 | 3e-1 | 8e-4 | 6e-3 |
| Contexts | 1e-43 | 1e-143 | 4e-138 | 5e-126 |
| Linear Maps | 3e-153 | 5e-2 | 5e-153 | 4e-146 |

Table 4: *Generic vs. specific modules.* For a W+S+D run of MUiR, this table gives two-tailed $p$-values (Mann-Whitney) comparing generic vs. specific weight tensors over four statistics for each parameter group: modules, contexts, and the linear maps they generate. The generic tensors tend to have a much tighter distribution of parameters, indicative of better generalization: They must be applied in many situations with minimal disruption to overall network behavior.

single task learning (STL) on two out of three tasks. This result shows the value of the universal decomposition in Section 3.1, even when used with other DMTL approaches.

| Method | LeNet | Stacked LSTM | DeepBind |
|---|---|---|---|
| Single Task Learning | 21.46 | 135.03 | 0.1543 |
| Classical DMTL (e.g., [13, 19, 61]) | 21.09 | 145.88 | 0.1519 |
| Parallel Adapters [44] | 21.05 | 132.02 | 0.1600 |
| MUiR + Hierarchical Init. | **20.72** | **128.94** | **0.1465** |
| MUiR | **20.51** | **130.70** | **0.1464** |

Table 6: *Comparison across DMTL methods.* MUiR outperforms the other methods, even with hierarchical initialization.

| $c$ | LeNet | Stacked LSTM | DeepBind |
|---|---|---|---|
| 0 | 21.89 | 144.52 | 0.1508 |
| 1 | 21.80 | 140.94 | 0.1477 |
| 2 | **20.40** | 133.94 | 0.1504 |
| 4 | **20.51** | **130.70** | **0.1464** |
| 8 | 20.62 | **130.80** | **0.1468** |

Table 7: *Comparison across c.* Hypermodules ($c > 0$) are beneficial.

Next, the significance of the $\psi$ initialization method was tested, by initializing MUiR with the hierarchical alignment used by the other methods, instead of the disjoint initialization suggested by Theorem 3.2. This method (Table 6: MUiR+Hierarchical Init.) still outperforms the previous methods on all tasks, but may be better or worse than MUiR for a given task. This result confirms the value of MUiR as a framework, and suggests that more sophisticated initialization could be useful.

The importance of hypermodule context size $c$ was also tested. Comparisons were run with $c = 0$ (blocks shared exactly), $1, 2, 4$ (the default value), and $8$. The results confirm that location-specific contexts are critical to effective sharing, and that there is robustness to the value of $c$ (Table 7).

Finally, MUiR was tested when applied to a highly-tuned Wikitext-2 baseline: AWD-LSTM [35]. Experiments directly used the official AWD-LSTM training parameters, i.e., they are tuned to AWD-LSTM, not MUiR. MUiR parameters were exactly those used in the other cross-domain experiments. MUiR achieves performance comparable to STL, while reducing the number of LSTM parameters from 19.8M to 8.8M during optimization (Table 5). In addition, MUiR outperforms STL with

| Method | LSTM Parameters | Perplexity |
|---|---|---|
| STL | 8.8M | 73.64 |
| MUiR | 8.8M | 71.01 |
| STL | 19.8M | 69.94 |

Table 5: Results on Wikitext-2 with AWD-LSTM [35].

the same number of parameters (i.e., with a reduced LSTM hidden size). These results show that MUiR supports efficient parameter sharing, even when dropped off-the-shelf into highly-tuned setups. However, MUiR does not improve the perplexity of the best AWD-LSTM model. The challenge is that the key strengths of AWS-LSTM comes from its sophisticated training scheme, not its architecture. MUiR has unified diverse architectures; future work must unify diverse training schemes.

# 5 Discussion and Future Work

Given a set of deep learning problems defined by potentially disparate (architecture,task) pairs, MUiR shows that learned functionality can be effectively shared between them. As the first solution to this problem, MUiR takes advantage of existing DMTL approaches, but it is possible to improve it with more sophisticated and insightful methods in the future. Hypermodules are able to capture general functionality, but more involved factorizations could more easily exploit pseudo-task relationships [32, 57]. Similarly, the $(1 + \lambda)$-EA is simple and amenable to analysis, but more sophisticated optimization schemes [10, 47, 54] may be critical in scaling to more open-ended settings. In particular, the modularity of MUiR makes extensions to lifelong learning [1, 6, 43, 52] especially promising: It should be possible to collect and refine a compact set of modules that are assembled in new ways to solve future tasks as they appear, seamlessly integrating new architectural methodologies. Such functionality is fundamental to general problem solving, providing a foundation for integrating and extending knowledge across all behaviors during the lifetime of an intelligent agent.

# 6 Conclusion

To go beyond methodological sharing in deep learning, this paper introduced an approach to learning sharable functionality from a diverse set of problems. Training a set of (architecture,task) pairs is viewed as solving a set of related pseudo-tasks, whose relatedness can be exploited by optimizing a mapping between hypermodules and the pseudo-tasks they solve. By integrating knowledge in a modular fashion across diverse domains, the approach establishes a key ingredient for general problem solving systems in the future.

**Acknowledgments**

Many thanks to John Hawkins for introducing us to the CRISPR binding prediction problem and providing the data set. Thanks also to the reviewers for suggesting comparisons across framework design choices and other DMTL methods.

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
