[Supplementary Material]

# S Supplemental Material

## S.1 Proof of Theorem 3.1.

*The expected time of the decomposed K-valued (1+1)-EA is $O\big(\frac{KL(\log L - \log D)\log D}{D}\big)$ for linear $h_d$.*

*Proof.* The proof is a direct extension of the result for the non-decomposed binary-valued algorithm [53], which converges in $\Theta(L\log L)$ iterations with high probability. Following that proof exactly, but replacing binary variables to $K$-valued variables increases the convergence time to $\Theta(KL\log L)$. Then, each subproblem in the decomposed version converges in $\Theta(K\frac{L}{D}\log\frac{L}{D}) = \Theta(\frac{KL(\log L - \log D)}{D})$ time with high probability, that is, the CDF of the convergence of each instance is dominated by an exponential random variable with mean $\Theta(\frac{KL(\log L - \log D)}{D})$. The maximum of $D$ i.i.d. exponential random variables with mean $1/\rho$ is $H_D/\rho$, where $H_D = \Theta(\log D)$ is the $D$th harmonic number [14]. So, the expected convergence time of the entire algorithm is $O(\frac{KL(\log L - \log D)\log D}{D})$. $\qquad\square$

## S.2 Proof of Theorem 3.2.

*The expected time of the decomposed K-valued (1+1)-EA with pessimistic initialization and proportional sampling is $O(\log L)$, when $D = L$, and all $h_d$ are linear.*

*Proof.* Let $W_i$ be a variable tracking the number of locations whose module is wrong at iteration $i$. $W_0 = L - 1$, since the first location is initialized correctly. Let $W_{i+1}$ be the expected number of locations whose module is incorrect at time $i + 1$ given that $W_i$ are incorrect at time $i$. Then,

$$W_{i+1} = W_i\big(1 - \frac{L - W_i}{L}\big) = \frac{W_i^2}{L}, \tag{9}$$

which yields a closed form for $W_t$:

$$W_t = \frac{1}{L}(\dots(\frac{1}{L}(\frac{1}{L}(L-1)^2)^2)\dots)^2 = \frac{(L-1)^{2^t}}{L^{2^t - 1}}. \tag{10}$$

If at most 1 location is incorrect, optimizing this location takes constant time. The goal is to find $t$ such that $W_t < 1$:

$$W_t = \frac{(L-1)^{2^t}}{L^{2^t - 1}} = L(\frac{L-1}{L})^{2^t} < 1$$

$$\implies 2^t < \log_{\frac{L-1}{L}}\frac{1}{L} = \frac{\ln\frac{1}{L}}{\ln\frac{L-1}{L}} < \frac{\ln\frac{1}{L}}{-\frac{1}{L-1}} = L\ln L - \ln L$$

$$\implies t < \log(L\ln L) = O(\log L). \tag{11}$$

Since the expected time to get from $W_i$ to $W_{i+1}$ is one iteration, and convergence is faster when $W_i$ is lower, $t$ is an upper bound on the expected runtime of the algorithm. $\qquad\square$

## S.3 Additional algorithm details

For the model to learn its hypermodule preferences efficiently, a special learning rate $\mathrm{lr}_s$ is assigned to the soft weights $s_\ell$ in Eq. 8. In the experiments, setting this rate to one or two orders of magnitudes larger than that of the rest of the model yields reliable results.

The complete end-to-end algorithm is given in Algorithm 2. The algorithm interleaves model training with optimization of $\psi$. Interleaving makes the algorithm efficient, because the model need not be trained from scratch each generation. Instead, $\lambda$ hypermodule options are sampled for each of $\lceil pL \rceil$ pseudo-tasks, for some $p \in (0, 1]$. Although in theory $p = 1$ yields the fastest convergence, setting $p < 1$ improves the stability of training, reducing the noise that comes from shocking pseudo-tasks with new modules. In the experiments, $p = 0.5$ was found to yield reliable results. Training can also be made smoother by training for $n_{\text{init}}$ steps before optimizing $\psi$, and by initializing the probability of the current best hypermodule to be $1 - \alpha$ for some small $\alpha < 1$. If $s_{\ell 0}$ is initialized to 0, then, for $i \in \{1, \dots, \lambda\}$

$$\text{softmax}(s_\ell)_i = \frac{\alpha}{\lambda} \implies s_{\ell i} = \ln\alpha - \ln\lambda - \ln(1 - \alpha). \tag{12}$$

---

**Algorithm 2** Interleaved optimization of module alignment

---
1: **Initialize** any non-sharable model parameters $\theta'$.
2: **Initialize** $\{H_\ell\}_{\ell=1}^{L}$, $\{z_\ell\}_{\ell=1}^{L}$, and $\psi^0$ with $\psi^0(\ell) = H_\ell$.
3: **Train** $H, z, \theta'$ via Eq. 5 for $n_{\text{init}}$ backprop steps.
4: **for** $n_{\text{gen}}$ generations **do**
5:      **for** $\ell = 1, \ldots, L$ **do**
6:          $s_{\ell 0} \leftarrow 0$
7:          $\psi^i \leftarrow \psi^0$
8:          **for** $i = 1, \ldots \lambda$ **do**
9:              $s_{\ell i} \leftarrow \ln \alpha - \ln \lambda - \ln(1 - \alpha)$
10:      $loc \leftarrow \lceil pL \rceil$-element random subset of $\{1, \ldots, L\}$
11:      **for** $\ell \in loc$ **do**
12:          **for** $i = 1, \ldots \lambda$ **do**
13:              $\psi^i(\ell) \leftarrow \text{randomHypermodule}(\psi^0)$
14:      **Train** $H, z, \theta', s$ via Eq. 8 for $n_{\text{iter}}$ backprop steps.
15:      **Evaluate** using the validation set for each task.
16:      **for** $\ell = 1, \ldots, L$ **do**
17:          $\psi^0(\ell) \leftarrow \psi^{\arg\max_i \sum_{j=0}^{\lambda} \text{softmax}(s_\ell)_j \cdot I(\psi_\ell^j = \psi_\ell^i)}(\ell)$
18: **Revert** to the state with best validation performance.
19: **Train** $H, z, \theta'$ via Eq. 5 for $n_{\text{final}}$ backprop steps.

---

However, in this paper, $\alpha = \frac{\lambda}{\lambda+1}$ in all experiments, so that there is no initial bias towards previously selected hypermodules.

Note that the choice of $\lambda$ is limited by scalability concerns. The cost of one gradient update is approximately $1 + \lambda$ times that of the original model. This pressure towards small $\lambda$ is why $\lambda = 1$ was used in Section 4.2. This scalability pressure also makes it crucial that the results in Section 3.3 apply in the case of $\lambda = 1$.

As required by Theorem 3.2, new hypermodules for a pseudo-task are selected with probability proportional to their current usage. When a hypermodule is no longer used anywhere, it has effectively been deleted. When the number of active hypermodules is less than the initial number $K$, for theoretical robustness, a small probability $\epsilon$ of creating a new hypermodule is always included, similar to the $\epsilon$-greedy approach in reinforcement learning [50]. In this paper, $\epsilon$ is manually set to $10^{-4}$ in all experiments. The distribution for sampling existing hypermodules is then

$$P(H_k|\psi^0) = \frac{(1-\epsilon)}{L}\big|\{\ell : \psi^0(\ell) = H_k\}\big|. \tag{13}$$

In practice, there may be some parameters that are not naturally decomposable via Eq. 2. In particular, the initial layer that transforms raw input and the output layer that produces predictions are modality-specific. They are useful as unshared *adapters* that learn permutations and scaling to translate between specific and generic representations. For example, for each task in S.5, the first and last layers of its architecture are reserved as adapters.

## S.4 Additional experiment details

All models were implemented in PyTorch [41]. Code is available at `github.com/leaf-ai/muir`. Each run was performed using a single NVIDIA GTX 1080 Ti GPU with 12GB RAM.

All models (except AWD-LSTM models) were trained using Adam with default parameters [24]. When the learned parameters $s_\ell$ are reset each generation, their corresponding auxiliary state in Adam is reset as well, to prevent unmeaningful application of this state.

The synthetic dataset in Section 4.1 contains 30 linear regression tasks, each with the same 20-dimensional input space and 1-dimensional output [23]. Each task was generated from a random parameter vector, by multiplying random inputs by this vector to generate 15 training samples and 50 test samples. The goal is to minimize RMSE averaged over all tasks. The tasks are grouped into three groups of ten tasks each. The parameter vector for tasks within a group differ only by a scalar factor. Tasks cannot be solved reliably without exploiting this regularity. The linear models in these experiments use a batch size of 10 in training.

For the results in Table 2, each setup was run ten times. Mean and standard error are reported. Surprisingly, in the clean case, the MUiR + Oracle setup performs worse than MUiR + Optimization. This result is due to the fact that the Oracle setup is still able to occasionally overfit to one of the thirty tasks, because there is so little data, and there are no other forms of regularization. In particular, note that the median RMSE for both MUiR + Oracle and MUiR + Optimization was 0.00. In the noisy case, the noise itself provides sufficient regularization for the Oracle to overcome this issue. However, the improvement of Optimization over Oracle in the clean case illustrates a strength of MUiR that is also captured in Table 4: Since each module is trained in many locations over the course of optimization, it is forced to learn generalizable functionality.

In Figure 2, the first 10 tasks correspond to the first ground truth group, the second 10 to the second group, and the third to the third group. The "Score" at each generation is a coarse measure for how close $\psi$ is to the optimal mapping: Each task adds 1 if the module it uses is shared and only used by tasks in its true group, adds 0 if the module is unshared, and adds -1 if the module is shared by tasks outside of its true group.

In the experiments in Section 4.1, 99 iterations of random search were performed for the noisy case over the hyperparameter ranges $\lambda \in \{1, 2, 4, 8\}$, $p \in \{0.1, 0.25, 0.5, 1\}$, $\mathrm{lr}_s \in \{0.01, 0.1, 1, 10\}$, and $n_{\mathrm{iter}} \in \{10, 50, 100, 200\}$. The setting with the best validation loss was $\lambda = 8$, $p = 0.5$, $\mathrm{lr}_s = 0.01$, and $n_{\mathrm{iter}} = 100$. This setting was then used across ten runs in both the clean and the noisy case to collect the results in Table 2. Since the linear models learn quickly, $n_{\mathrm{init}}$ was not needed and set to 0.

To scale up to the experiments in Section 4.2, the hyperparameter settings above were copied exactly, except for $\lambda$, $\mathrm{lr}_s$, $n_{\mathrm{iter}}$, and $n_{\mathrm{init}}$, which were manually adapted from those in Section 4.1: $\lambda$ was set to 1 for maximum computational efficiency; $\mathrm{lr}_s$ was increased to 0.1 so that locations could quickly ignore clearly low-performing modules; $n_{\mathrm{iter}}$ was increased to 1000 to handle the larger problem size; $n_{\mathrm{init}}$ was set to 2000 so that model could initially stabilize before alignment optimization.

In Section 4.2, one run was performed for each of the setups in Table 3, i.e., five to seven runs were performed for each architecture. To confirm the significance of the results, twenty additional runs were performed for the baselines L, S, and, D, as well as for the cross-domain setup L+S+D. The means are shown in Table 3. The mean ($\pm$ std. err.) for the baselines was 21.08 ($\pm 0.09$), 0.1540 ($\pm 0.0005$), and 134.41 ($\pm 0.62$), respectively, while for L+S+D they were 20.23 ($\pm 0.08$), 0.1464 ($\pm 0.0002$), and 130.77 ($\pm 0.12$). For all three of these improvements $p < 1\mathrm{e}^{-4}$ (Welch's t-test).

In the results in Table 4 there were 666 generic modules, 4344 specific; and 4363 generic pseudo-tasks (i.e., contexts and linear maps) and 8401 specific. Notably, the differences between generic and specific tensors appear for both hypermodules, which are trained for a variable number of pseudo-tasks, and contexts, which are each trained for only one pseudo-task.

For computational constraints, experiments in Section 4.2.3 were capped at 200 epochs. Parallel adapters were implemented by including a location-specific learned diagonal matrix for each pseudo-task location, which is added to the parameter block used at that location in order to adapt it. AWD-LSTM experiments were implemented based on the official AWD-LSTM implementation: https://github.com/salesforce/awd-lstm-lm. When applied to AWD-LSTM, MUiR jointly trained AWD-LSTM and LeNet, since L+S performs best on Wikitext-2 in Section 4.2.1.

## S.5 Dataset and architecture details

*CIFAR-10.* This image classification benchmark has 50,000 training images and 10,000 test images [26]. Of the training images, 5,000 are randomly withheld for validation. As in previous work on hypernetworks, WideResNet-40-1 (WRN) is the underlying model, and standard data augmentation is used [15]. The first and last layers of the model are reserved as adapter layers. All remaining convolutional layers are reparameterized by hypermodules, yielding a total of 2268 blocks. WideResNet defines a family of vision models, each defined by a depth parameter $N$ and a width parameter $k$. WideResNet-40-1 has $N = 6$ and $k = 1$. This model is the smallest (in terms of parameters) high-performing model in the standard WideResNet family. For the additional set of experiments using LeNet [29] as the vision model, all layer sizes were increased to the nearest multiple of 16. This model is sequential with five layers, of which the middle three are reparameterized. Both CIFAR-10 models use a batch size of 128 for training.

*WikiText-2.* This language modeling benchmark has 2,088,628 training tokens, 217,646 validation tokens, and 245,569 test tokens, with a vocab size of 33,278 [36]. The goal is to minimize perplexity. The underlying model is the standard stacked LSTM model with two LSTM layers each with 256 units, and preprocessing is performed as in previous work [59]. The LSTM layers are reparameterized by hypermodules, yielding a total of 4096 blocks. This standard model has one main parameter, LSTM size. In general, increasing the size improves performance. Common LSTM sizes are 200, 650, and 1000. To simplify the setup by making the LSTM weight kernels divisible by the output dimension of hypermodules, the experiments in Section 4.2 use an LSTM size of 256. The model begins with a word embedding layer, and ends with a dense layer mapping its output to a softmax over the vocabulary. This model uses a batch size of 20 for training.

*CRISPR Binding Prediction.* The goal of this dataset is to predict the propensity of a CRISPR protein complex to bind to (and cut) unintended locations in the genome [21]. This is an important personalized medicine problem, since it indicates the risk of the technology for a particular genome. When using the technology, there is one particular (target) location that is intended to be cut out by the CRISPR complex, so that this location can be edited. If the complex makes other (off-target) cuts, there may be unintended consequences. Predicting the binding affinity at off-target locations gives an assessment of the risk of the procedure. The dataset contains binding affinities for approximately 30 million base pairs (bp). Input consists of 201bp windows of one-hot-encoded nucleobases centered around each location. The data is randomly split into non-overlapping training, validation, and test sets, with approximately one million samples withheld for validation and one million for testing. The underlying model, DeepBind-256, is from the DeepBind family of 1D-convolutional models designed for protein binding problems [2, 60]. The first layer embeds the input into 256 channels. The second layer is a 1D convolution with kernel size 24, and 256 output channels, followed by global max pooling. The third layer is fully-connected with 256 hidden units. The final layer is fully-connected with a single output that indicates the predicted binding affinity. The loss is MSE. The middle two layers are reparameterized by hypermodules, yielding 6400 blocks. This model uses a batch size of 256 for training.