[Reviews · NeurIPS 2019]

Reviewer 1



Edit: Thanks to the authors for addressing my comments and running additional experiments. I have consequently increased my score. The paper proposes to decompose the parameters into L distinct parameter blocks. Each of these blocks is seen as solving a "pseudo-task", learning a linear map from inputs to outputs. The parameters of these blocks are generated by K hypermodules (small hypernetworks) that condition on a context vector for each pseudo-task based. The alignment of hypermodules to pseudo-tasks is governed by a softmax function and learned during training similar to mixture-of-experts. By sampling hypermodules proportional to their usage, more general modules are used more often. The approach is evaluated on a synthetic dataset (modeling linear regression tasks) and on a setting that involves three cross-modal tasks: object recognition on CIFAR-10, language modeling on WikiText-2, and CRISPR binding prediction. The approach combines hypernetworks and mixture-of-experts in an interesting way to learn to capture information that can be shared across different tasks. It is interesting to see that some hyper-modules become general and are used many pseudo-tasks. My main concerns about this submission regard its evaluation and lack of analysis of model components. The main experiment of the paper is the evaluation on cross-modal multi-task learning using a vision, a text, and a DNA task. However, based on the provided scores, the actual performance of the model remains unclear as the results are a) not competitive and b) not compared to any multi-task learning baselines. a) is most apparent for language modelling (which I am most familiar with), where the state-of-the-art on WikiText-2 is around 39 perplexity (lower is better) and reasonably tuned models from 2017 (https://arxiv.org/abs/1708.02182) achieve around 65 perplexity—compared to 128 reported here. For CIFAR-10, there is a (albeit smaller) gap to the results from the WideResNet or Hypernetworks papers that the method uses. State-of-the-art performance importantly is not necessary, but in order to make a statement about any gains from the new model, the proposed method should at least be competitive with results from the literature. Re b), I am missing a comparison against other multi-task learning models. Even if these are found not to be feasible in the cross-modal setting, a comparison would still be useful to demonstrate how challenging the benchmark is. In addition, the proposed model should be compared to other cross-modal MTL models such as the MultiModel (https://arxiv.org/abs/1706.05137). Alternatively, to get more confidence in the performance of the model, I would have appreciated seeing a comparison on a standard MTL benchmark of a single modality. Finally, as the model involves many subjective choices regarding the number of pseudo-tasks, the number of hyper-modules, the initialization and sampling strategies, etc. I would have liked to see an ablation analysis that studies and identifies the most impactful components of the model. Overall, even though I like the proposed approach, I do not think it is ready for acceptance at this point and think that it could be significantly improved through more careful experiments and comparison to prior work.

Reviewer 2



The paper is well written. The problem is well described and formalized. In some points the theory is a bit hard to follow even becouse some proof/explanation are postponed to the supplementary, but this is of course due to the space limits. The aim of having 'one' deep architecture that can accomodate multiple learning tasks which benefit from reciprocal transfer even beyond their specific differences is challening and of large interest for the community.

Reviewer 3



The content of this paper is dense at times but the clarity and writing quality is good for the most part. The main contribution is quite interesting as the paper proposes a new framework MUiR that can share parameters across different tasks even when the architectures used are different. Nice analysis of the algorithm time and complexity are also provided in section 3. Where the paper falls short for me is in the theoretical and empirical comparisons with prior work. For the synthetic results MUiR is compared to some multi-task learning baselines. The experiments seem to indicate that other algorithms can achieve similar or better performance. For the cross-modal results MUiR achieves good performance, but is not compared to any relevant architecture selection models. I still lean towards accept because of the nice framework and algorithmic analysis. However, the paper could be significantly improved by providing some more competitive multi-task learning baselines as part of the the cross-modal results for context.

[Author Response · NeurIPS 2019]

Thank you to the reviewers for the feedback. There were three main suggestions: additional comparison to other approaches,
experiments varying the settings of MUiR, and experiments with a highly-tuned baseline. These concerns are addressed below with
additional experiments that were run for this rebuttal and confirm the advantages of our approach. These experiments are described
in the first three paragraph below, and are complemented by responses to specific reviewer comments in the rest of the response.

**Reviewers 2 and 4** were interested in comparisons to other deep multi-task learning (DMTL) methods in the cross-modality problem
in this paper, even if their application seems unnatural for this problem. So, experiments in the three-domain setting were run
using *classical* DMTL (e.g., [1, 2, 10]), i.e., where aligned parameters are shared exactly across tasks, and *parallel adapters* [8], an
approach mentioned by **Reviewer 3**, which is state-of-the-art (SoA) for vision MTL. Both of these methods require a hierarchical
alignment of parameters across architectures. Here, the most natural hierarchical alignment is used, based on a topological sort of the
block locations within each architecture: the $i$th location uses the $i$th parameter block. MUiR outperforms the existing methods
(Table 1). Interestingly, the existing methods each outperforms single task learning (STL) on two out of three tasks. This result
shows the value of the universal decomposition in Section 3.1, even when used with other DMTL approaches.

As suggested by **Reviewer 2**, an additional experiment of MUiR was run with different initialization. In this experiment, the module
mapping is initialized with the hierarchical alignment used by the other methods above, instead of using the separate initialization
suggested by the theory in the paper. This method (Table 1: MUiR+Hierarchical Init.) still outperforms the previous methods on
all tasks, but may be better or worse than MUiR for a given task. This result confirms the value of MUiR as a framework, and
indicates that exploring initialization schemes is a promising area of future work. Additional exploration of experimental settings
will be included in the final version of the paper. The design decisions of MUiR were intended to be the simplest solutions given the
requirements of the theory, and it is expected that many such future innovations are possible that would improve the system.

**Reviewer 2** was interested in how MUiR would perform in recent highly-tuned training setups for Wikitext-2, e.g., AWD-LSTM [5].
Experiments were run based on the official AWD-LSTM implementation, directly using the many training parameters provided
there, i.e., they are tuned to AWD-LSTM, not MUiR. The parameters of MUiR were exactly those used in the other cross-domain
experiments. Table 2 shows the results. MUiR achieves performance comparable to STL, while reducing the number of LSTM
parameters from 19.8M to 8.8M during optimization. In addition, MUiR outperforms STL with the same number of parameters
(i.e., with a reduced LSTM hidden size). These results show that MUiR supports efficient parameter sharing, even when dropped
off-the-shelf into highly-tuned setups. The final version of the paper will include results on AWD-LSTM with tuning of MUiR.

Table 1: Comparison to other methods and alternative initialization.

| Method | LeNet | Stacked LSTM | DeepBind |
|---|---|---|---|
| Single Task Learning | 21.46 | 135.03 | 0.1543 |
| Classical DMTL (e.g., [1, 2, 10]) | 21.09 | 145.88 | 0.1519 |
| Parallel Adapters [8] | 21.05 | 132.02 | 0.1600 |
| MUiR + Hierarchical Init. | **20.72** | **128.94** | **0.1465** |
| MUiR | **20.51** | **130.70** | **0.1464** |

Table 2: Results on Wikitext-2 with AWD-LSTM [5].

| Method | LSTM Parameters | Perplexity |
|---|---|---|
| Single Task Learning | 8.8M | 73.64 |
| MUiR | 8.8M | 71.01 |
| Single Task Learning | 19.8M | 69.94 |

**Reviewer 2** was interested in a comparison to MultiModel [3]. MultiModel does not address the question of how to parameterize a
given set of architectures. It also has gaps with the SoA, and only reports a subset of performance results, which is understandable,
since the cross-domain problem is so challenging (as **Reviewer 3** notes). Overall, MultiModel seems like a promising orthogonal
direction, and a comparison is not relevant at this time, though it may be relevant in the future if the approaches converge.

**Reviewer 3** asked about connections to sequential and parallel adapters [8]: As Table 1 shows, the value of such methods can
generalize beyond vision, although they are quite compact, and not theoretically as flexible as hypermodules. **Reviewer 3** asked
about more details for when tasks must be learned in a sequence [7]: The most natural approach is to initialize the module set for a
new task with existing modules, coupled with a method for preventing forgetting. We will expand on these points in the final version.

Beyond the experimental comparisons above, **Reviewer 4** asked about other theoretical comparisons to previous work on automatic
design of MTL models. We will expand on the following in the paper: Learning the alignment with soft ordering [6] yields a
quadratic increase in module operations, which is infeasible; Sampling from the softmax instead would still require thousands
of additional parameters per module location; The complexity of CTR [4] is shown to be infeasible via Theorem 3.1; Existing
approaches use at most 4 [6], 4 [4], and 10 [9] modules, resp., several orders of magnitude fewer than what is considered here, i.e.,
the cross-domain experiments in the paper use more than 10K modules, and the AWD-LSTM experiment uses more than 60K.

[1] D. Dong, H. Wu, H. We, D. Yu, and H. Wang. Multi-task learning for multiple language translation. In *Proc. of ACL*, pages 1723–1732, 2015.
[2] Z. Huang, J. Li, S. M. Siniscalchi, et al. Rapid adaptation for deep neural networks through multi-task learning. In *Proc. of Interspeech*, pages 3625–3629, 2015.
[3] L. Kaiser, A. N. Gomez, N. Shazeer, A. Vaswani, N. Parmar, L. Jones, and J. Uszkoreit. One model to learn them all. *CoRR*, abs/1706.05137, 2017.
[4] J. Liang, E. Meyerson, and R. Miikkulainen. Evolutionary architecture search for deep multitask networks. In *Proc. of GECCO*, 2018.
[5] S. Merity, N. S. Keskar, and R. Socher. Regularizing and optimizing LSTM language models. In *Proc. of ICLR*, 2018.
[6] E. Meyerson and R. Miikkulainen. Beyond shared hierarchies: Deep multitask learning through soft layer ordering. In *Proc. of ICLR*, 2018.
[7] S.-A. Rebuffi, H. Bilen, and A. Vedaldi. Learning multiple visual domains with residual adapters. In *NIPS*, pages 506–516. 2017.
[8] S.-A. Rebuffi, H. Bilen, and A. Vedaldi. Efficient parametrization of multi-domain deep neural networks. In *Proc. of CVPR*, pages 8119–8127, 2018.
[9] C. Rosenbaum, T. Klinger, and M. Riemer. Routing networks: Adaptive selection of non-linear functions for multi-task learning. In *Proc. of ICLR*, 2018.
[10] Z. Zhang, L. Ping, L. C. Chen, and T. Xiaoou. Facial landmark detection by deep multi-task learning. In *Proc. of ECCV*, pages 94–108, 2014.

* For time constraints, all experimental results in tables were capped to 200 epochs.

[Meta-Review · NeurIPS 2019]

The submission is proposing a multi-task learning method based on sharing linear submodules. The proposed idea is interesting, novel, and shown to be effective. On the other hand, reviewers raised various issues about the empirical study. Authors did a good job addressing this issue in their response, and the final evaluation of all reviewers are positive. The paper is a good addition to the conference, and I recommend acceptance. Authors should add the promised experimental results in the camera-ready version.